# Sexual Dimorphism in the Mechanism of Pain Central Sensitization

**DOI:** 10.3390/cells12162028

**Published:** 2023-08-09

**Authors:** Ellane Barcelon, Seohyun Chung, Jaesung Lee, Sung Joong Lee

**Affiliations:** 1Department of Physiology and Neuroscience, School of Dentistry, Dental Research Institute, Seoul National University, Seoul 08826, Republic of Korea; ellanebarcelon@snu.ac.kr (E.B.); sh113@snu.ac.kr (S.C.); lddlwotjd@snu.ac.kr (J.L.); 2Department of Interdisciplinary Program in Neuroscience, College of Natural Sciences, Seoul National University, Seoul 08826, Republic of Korea

**Keywords:** central sensitization, neuropathic pain, sexual dimorphism, microglia

## Abstract

It has long been recognized that men and women have different degrees of susceptibility to chronic pain. Greater recognition of the sexual dimorphism in chronic pain has resulted in increasing numbers of both clinical and preclinical studies that have identified factors and mechanisms underlying sex differences in pain sensitization. Here, we review sexually dimorphic pain phenotypes in various research animal models and factors involved in the sex difference in pain phenotypes. We further discuss putative mechanisms for the sexual dimorphism in pain sensitization, which involves sex hormones, spinal cord microglia, and peripheral immune cells. Elucidating the sexually dimorphic mechanism of pain sensitization may provide important clinical implications and aid the development of sex-specific therapeutic strategies to treat chronic pain.

## 1. Introduction

Sexual dimorphism has received attention in recent decades as a result of numerous studies that have identified sex differences not only in clinical studies, but also in basic and animal studies of neurological diseases, particularly in the study of pain. The International Association for the Study of Pain defined pain as “an unpleasant sensory and emotional experience associated with actual or potential damage” [1]. In contrast to acute pain, which resolves when the tissue injury heals, chronic pain persists beyond the point of wound healing, leading to pathological changes in the peripheral and/or central nervous system [2,3,4]. Despite the debilitating effects of chronic pain, effective treatments are currently limited. Some of the most evident limitations of pathological pain treatment are the partial efficacy and negative side effects of clinically approved medicines in subpopulations of patients [5,6,7,8]. Studies indicate the existence of sexual dimorphism in chronic pain, with women showing greater susceptibility to pain than men in the majority of chronic pain conditions [6,9,10,11,12]. Several drugs for chronic pain treatment also showed mechanism differences in males and females [13,14]. Yet, in most basic studies of chronic pain using animal models, sex has not been regarded as a biological variable, and preclinical research has used mostly male animals. As the study of pain and its pathology continues to expand, recent studies investigating female animals have revealed large sex differences in areas ranging from gene-expression profiles to animal behaviors in pain sensitization [15]. These recent studies clearly showed the importance of studying sexual dimorphism in pain and other neurological disorders for development of effective sex-specific therapeutic strategies. In this study, we review recent findings from animal research on the sexual dimorphism of pain central sensitization, in particular discussing putative molecular and cellular mechanisms.

## 2. Pain Central Sensitization

Pain central sensitization normally refers to a state of increased reactivity of pain-transmitting neurons or circuits in the central nervous system (CNS), resulting in greater sensitivity to nociceptive stimuli and elicitation of pain signals against non-noxious stimuli [16]. Central pain sensitization can develop through an increase of excitability and/or synaptic efficacy of nociceptive neural circuits or a reduction of inhibitory synaptic inputs in the nociceptive pathways at the spinal cord and supra-spinal cord levels. Activation of postsynaptic glutamate NMDA receptors, recruitment of AMPA receptors into the post-synaptic membrane, and alteration of chloride transporter KCC2 expression have been proposed as putative mechanisms underlying such increased pain synaptic activity at the spinal cord level [17,18]. Molecular factors inducing such alterations include glutamate, substance P, and CGRP secreted from primary afferents along with BDNF, interleukin (IL)-1β, and tumor necrosis factor (TNF)-α from activated spinal cord microglia [19,20].

Pain central sensitization manifests as both inflammatory nociceptive pain and neuropathic pain. Inflammatory nociceptive pain is a stimulus-dependent pain that occurs when nociceptors detect noxious or potentially harmful stimuli. Inflammation is an essential physiological process following injury or infection in the body that recruits inflammatory leukocytes, such as neutrophils and macrophages, to affected sites, triggering tissue repair and healing. In this process, inflammatory cells release molecules such as bradykinins that can directly activate nociceptive sensory neurons [21]. The information encoded as electrical signals in nociceptive neurons is transmitted to pain projection neurons in the spinal cord via afferent nerves. During this process, various biological mediators in addition to neurotransmitters (e.g., BDNF, CGRP, substance P, chemokines, and cytokines) are released, contributing to the sensitization of neural circuits at the spinal cord level [22,23,24]. In addition, damage or dysfunction in the nervous system often results in pain central sensitization, which is referred to as neuropathic pain. Clinical symptoms of neuropathic pain include spontaneous burning sensation, allodynia, and hyperalgesia. Such symptoms can be caused by central sensitization of the somatosensory nervous system in the aftermath of peripheral nerve injury, stroke, or spinal cord injury [25]. Not all peripheral or central nervous injuries lead to neuropathic pain, but up to 60% of patients with severe clinical neuropathy experience neuropathic pain [26], indicating it as a major health problem that reduces quality of life and often is accompanied by sleep disturbances, anxiety, and depression [27,28,29]. Recently, many clinical and preclinical studies have indicated sexual dimorphism in pain central sensitization.

## 3. Sexual Dimorphism of Pain in Clinical Studies

Human studies suggest that several (sociocultural, psychological, and biological) factors underlie the disparities in pain symptoms and response to pain treatments in men and women. One of the most focused topics is the influence of sex hormones on pain sensitivity [30,31,32]. Although existing studies are limited, testosterone is suggested to be anti-nociceptive and protective [33]. Meanwhile, estradiol and progesterone were reported to elicit both pro- and anti-nociceptive effects. Women with high estradiol levels exhibit decreased pain sensitivity and increased effectivity to analgesics compared to women with low estradiol levels [34]. A previous report also suggested that pain sensitivity in women changes throughout the menstrual cycle, with heightened sensitivity during the luteal phase [35]. These findings suggest the critical roles of sex hormones as determinants of sex-based disparities in pain sensitization.

Another factor being considered in sex differences in pain sensitization is age. Most preclinical and clinical pain research studies have enrolled primarily adult subjects. However, pain sensitivity to noxious stimuli changes during the lifespan [36,37,38]. It was reported that girls (6–8 years old) have lower pain thresholds to temperature and pressure than boys, but no difference was found in the mechanical pain threshold between the sexes at this age range [39]. Pain threshold and severity significantly vary between girls and boys, and the difference is most prominent at a mean age >12 years [36]. Hence, it was suggested that pubertal development and sex hormones impact these disparities in pain response between adolescent boys and girls [40].

## 4. Sexual Dimorphism of Pain in Pre-Clinical Studies

Preclinical studies primarily use various animal pain models, focusing on discrete peripheral nerve or tissue injuries that result in inflammatory and/or neuropathic pain. The tractability of animal pain neural circuits beginning from the site of the injury, through the dorsal root ganglia (DRG), dorsal horn of the spinal cord, and up to the brain has allowed researchers to investigate cellular and molecular cues that produce and regulate pain central sensitization. Here, we describe several animal pain models that exhibited sexually dimorphic pain behaviors and factors that have been discovered and are currently being investigated to elucidate the underlying mechanisms of sexually dimorphic pain central sensitization.

### 4.1. Sexual Dimorphic Pain Responses in Various Animal Models

Spared nerve injury (SNI) is one of the most used neuropathic pain models and is a partial denervation injury of the common peroneal and tibial nerves. SNI produces consistent and tactile hypersensitivity in the skin area of the spared, intact sural nerve [41]. Such pain symptoms have been induced by SNI in both male and female mice. It was reported that metformin, an inhibitor of protein kinase A, exerts analgesic effects on SNI-injured male mice; however, such analgesic effects were not observed in similarly treated female mice [42]. Similarly, although both SNI-injured male and female mice manifested mechanical allodynia, only male mice showed reversal of allodynia upon administration of minocycline (a microglia inhibitor), MAC-1-conjugated saporin (used to deplete microglia), or TNP-ATP (a P2X inhibitor and regulator of microglial activation) [43]. These studies indicate that the neuropathic pain mechanism due to SNI is distinct between male and female mice. 

Spinal nerve ligation (SNL), another model of neuropathic pain, is developed by tightly ligating the lumbar segment spinal nerve (L5 or L5 and L6) [44]. This causes long-lasting mechanical allodynia, hyperalgesia, and spontaneous pain. RNA sequencing data from DRG sensory neurons of both male and female mice with SNL showed comparable gene-expression profile changes upon nerve injury. There was no significant difference in terms of immune cell activation in the spinal cord; SNL increased immune cell infiltration and microglia proliferation in the spinal cord of both male and female mice at 24 days post-SNL. Notably, a sex difference was observed in immune cell infiltration into the DRG 8 days after SNL; B-cell infiltration in DRG was increased in male mice, while T-cells showed greater infiltration in female mice [45]. These findings revealed a sexually dimorphic adaptive immune response, especially in the DRG, upon SNL.

The sexual dimorphism of pain is also observable in chronic constriction injury (CCI) models of neuropathic pain. CCI involves loosely tying three or four constricting chromic ligatures around the sciatic nerve, causing Wallerian degeneration of axons [46]. Male and female mice with CCI exhibited sexually dimorphic gene-expression profiles in DRG. RNA sequencing data from DRG found a total of 146 genes (e.g., *Ahr*, *Fos*, *Socs3*) upregulated in CCI-induced female mice but not in male mice, whereas a total of 859 genes (e.g., *Kcna4*, *Cacna2d3*, *Scn8a*) were upregulated in CCI-induced male mice but not female ones [15]. In another study, CCI-induced male mice exhibited a gradual decrease in allodynia and complete recovery, while allodynia and gliosis persisted for >4 months after CCI surgery in female mice. Interestingly, estrogen administration in the CCI-induced female mice significantly attenuated allodynia and led to complete recovery within 13 weeks after administration, implicating the influence of estrogen in allodynia in female mice [47].

Lipopolysaccharide (LPS) is a type of endotoxin found in the outer membrane of Gram-negative bacteria that induces neuroinflammation upon intrathecal administration to an experimental subject. At the spinal cord, LPS activates microglia through toll-like receptor 4 (TLR4), inducing pro-inflammatory gene expression [48,49]. Interestingly, intrathecal administration of LPS in the spinal cord induced mechanical allodynia only in male but not female mice. On the contrary, LPS administration to the brain and hindpaw produced comparable degrees of pain hypersensitivity in mice of the two sexes [49]. The sexual dimorphism in LPS-induced pain sensitization is also associated with sex hormones; orchiectomized male or testosterone-deficient male mice exhibited reduced allodynia upon intrathecal LPS administration, whereas testosterone replacement reinstated the pain central sensitization [49,50], indicating that male sex hormones are involved in the sexual dimorphism. Notably, LPS stimulation led to sexually dimorphic pro-inflammatory cytokine production in primary astrocytes; TNF-α level was higher in males, while IL-10 level was higher in females [51].

Moreover, chronic pain has also been investigated in metabolic disorder neuropathies [52,53]. Preclinical models of chronic pain induced by metabolic disorder neuropathies are based on chemical and surgical approaches, genetic, and diet alterations. A recent study showed that sexual dimorphism has also been observed in metabolic disorders associated with chronic pain development [54]. The study showed that male and female mice with vitamin D deficiency were more susceptible to allodynia when compared to controls. Also, spinal cord and brain microglial proliferation was significantly higher in vitamin D-deficient females compared to male mice [54]. 

Sexual dimorphism in chronic pain was also observed in chemotherapy-induced peripheral neuropathy (CIPN) model [55]. CIPN is due to the nerve damage induced by chemotherapeutic agents. Studies showed that differences in males and females in chronic pain manifestation in CIPN depend on the chemotherapeutic agents being used [56]. Several chemotherapeutic agents produce CIPN through indirect or direct increasing of sensory neurons excitability. Moreover, different compounds of these chemotherapeutic agents act on various cellular and molecular targets inducing mitochondrial and DNA damage, increasing oxidative stress, and neuroinflammation, which all may lead to the sensitization of the peripheral nociceptor and sensory terminals. Studies suggest that potentiation of nociceptor channels such as the transient receptor potential channels (TRP) underlies the molecular mechanism involved in CIPN [55].

### 4.2. Factors of the Sexually Dimorphic Pain Mechanisms

Microglia, CNS resident immune cells, are the principal component of the sex differences in pain central sensitization [57]. In a resting state, microglia constantly surveil the CNS microenvironment. Upon tissue injury or infection, microglia are activated and change their morphology with increasing soma volume and decreasing process length and complexity. In addition, microglia robustly proliferate, release various bio-active diffusible factors like inflammatory cytokines and chemokines, and induce phagocytosis [58,59]. They are also involved in synaptogenesis and synaptic plasticity through interaction with neurons [60,61].

Microglia in the spinal dorsal horn proliferate robustly in response to peripheral nerve injury in both males and females. Peripheral nerve injury triggers spinal cord microglia activation, leading to p38 MAPK activation, P2X4R upregulation, and BDNF release, which results in allodynia in male mice. Therefore, depletion or inactivation of spinal cord microglia attenuates nerve injury-induced mechanical allodynia in male mice. For instance, intrathecal injection of minocycline elicited a dose-dependent reversal of allodynia after SNI in male mice [24]. Similarly, depletion of spinal cord microglia upon injection of MAC-1-conjugated saporin led to significantly reduced allodynia in male mice [43]. However, although microglial proliferation and morphological activation were observed upon peripheral nerve injury in female mice, depletion or inactivation of spinal cord microglia failed to rescue the female mice from the nerve injury-induced allodynia. Therefore, in female mice, nerve injury-induced pain central sensitization is independent of spinal cord microglia activation [43]. Studies on this sexually dimorphic mechanism of pain revealed that TLR4, a receptor uniquely expressed in microglia in the spinal cord, is required for pain hypersensitivity in male but not female mice. LPS activation of TLR4 in spinal microglia results in robust mechanical allodynia in male mice but not female ones [49]. 

In addition, studies reported that microglia–neuron communication via the P2X4R-p38 MAPK-BDNF pathway led to sexually dimorphic effects in chronic pain [62,63]. During peripheral nerve damage, P2X4R expression is increased in the spinal dorsal horn only in male mice. Microglial P2X4R is activated by ATP released by neurons in the spinal dorsal horn, activating microglia and producing hypersensitivity only in males [63,64,65,66]. Therefore, intrathecal administration of TNP-ATP, an inhibitor of P2X4R, attenuated SNI-induced allodynia only in male rats and mice [43,63]. This spinal microglial P2X4R activation by ATP leads to an influx of Ca^2+^ and p38 MAPK activation. A previous study reported that CCI induced p38 activation in spinal microglia in male but not female mice, although comparable microglia proliferation was found in both sexes [67]. CCI also enhanced the frequency of excitatory synaptic transmission of the spinal cord lamina IIo neurons in both sexes, which may have contributed to allodynia, but treatment of lumbar spinal cord slices with skepinone, a highly selective p38α inhibitor, suppressed the increased excitatory frequency of neurons only in male mice [67]. Likewise, intrathecal administration of skepinone [51] and another p38 MAPK inhibitor, SB 203580, reduced CCI- and SNI-induced allodynia in male mice but not female ones [43,67]. 

Studies showed that nerve injury-induced activation of microglial P2X4R and p38 MAPK leads to the synthesis and release of BDNF, which is necessary for the maintenance of pain hypersensitivity. Microglial release of BDNF acts through TrkB in dorsal horn neurons, causing a shift in the neuronal anion gradient underlying pain sensitization [18]. Intrathecal administration of BDNF inhibitor (Y1036) or BDNF-sequestering fusion protein (TrkB-Fc) reduced the SNI-induced allodynia in only male mice. Similarly, microglia-specific deletion of BDNF ameliorated pain in SNI-induced male mice but not female ones [43]. Thus, microglia-released BDNF is involved in the sexual dimorphism in pain central sensitization. 

Colony-stimulating factor 1 (CSF1) is also implicated in sexually dimorphic pain central sensitization. During peripheral nerve injury, sensory neurons upregulate CSF1, which is then transported and released in the spinal cord and induces microglial activation and proliferation, contributing to pain hypersensitivity [68]. Kuhn and colleagues discovered that, even in the absence of injury, intrathecal administration of CSF1 alone is enough to induce allodynia in male mice but not female ones [69]. CSF1 stimulated spinal cord microglia activation and subsequent induction of inflammatory gene transcription only in male mice. Notably, CSF1 administration in female mice induced remarkable infiltration of lymphocytes, preferentially regulatory T-cells, within the spinal cord meninges. Intriguingly, once regulatory T-cells were recruited to the female spinal cord, they suppressed CSF1-induced spinal cord microglia activation in female mice [69]. Thus, intrathecal CSF1 administration in regulatory T-cell-deficient female mice induced levels of microglial activation and allodynia comparable to those of male mice. These findings indicate that spinal cord microglia are activated via distinct pathways in male and female mice during CSF1-induced pain hypersensitivity. Moreover, these reports substantiate the idea that mechanisms of pain sensitization in males are solely dependent on microglia, while this may not be critical for pain central sensitization in females.

Studies show that female bodies utilize adaptive immune cells in addition to microglia to trigger pain central sensitization after nerve injury [43,69,70]. Specifically, in microglia-inactivated female mice, the infiltration of T-cells may contribute to the development of allodynia after peripheral nerve injury [69,71,72]. In a study using T-cell-deficient nude mice and recombination-activating gene 1 knockout (Rag1^−^/^−^) mice, a mouse line lacking mature T-cells and B-cells, microglia depletion or inactivation rendered female mice refractory to nerve injury-induced pain central sensitization. Therefore, female mice become sensitive to the analgesic efficacy of microglial inhibitors upon T-cell depletion [43]. Furthermore, studies indicate that the role of T-cells is regulated by peroxisome proliferator-activated receptors (PPARs), particularly PPARγ, which negatively regulates T-cell activation and decreases inflammatory cytokine expression by T-cells [73,74]. Thus, intrathecal administration of a PPARγ agonist, pioglitazone, attenuated SNI-induced hypersensitivity in female mice but not male ones [43]. These findings suggest that T-cells may be a critical component of the pain central sensitization mechanisms in females. 

Similar to CSF1 and LPS, we also found that GT1b, an endogenous agonist of TLR2 in the spinal cord [75], induces pain central sensitization in a sexually dimorphic manner. In our prior study, damaged sensory neurons produced GT1b, which was transported to the spinal cord and induced spinal cord microglia activation and central sensitization [75]. Intrathecal administration of GT1b induced spinal cord microglia activation with morphological changes such as enlarged cell bodies, microglial proliferation, and enhanced Iba-1 immunoreactivity. In addition, GT1b stimulation upregulated inflammatory cytokines such as IL-1β, TNF-α, and NADPH oxidase 2 in the spinal cord microglia [75,76]. Interestingly, although GT1b stimulation-evoked microglial activation and proliferation were observed in both male and female mice, pain sensitization was apparent only in the male mice [76]. We investigated the sexually dimorphic features of spinal cord microglia activation after GT1b administration by comparing gene-expression profiles of male and female mice. DAVID Gene Ontology analysis showed that most enriched differentially expressed genes in biological processes are associated with estrogen responses, including *Tph2*, *Krt19*, *Abcc2*, and *Agtr1b* genes (data are elaborated on by Lee and colleagues [76]).

We found that depleting systemic estrogen via bilateral ovariectomy rendered female mice susceptible to pain sensitization after GT1b administration, which was reversed by exogenous estrogen supplementation with 17β-estradiol. This finding showed that estrogen is responsible for sexually dimorphic GT1b-induced central pain sensitization. Meanwhile, orchiectomy-subjected male mice displayed similar levels of pain sensitivity to GT1b. This finding demonstrates that male sex hormones are not involved in GT1b-induced pain sexual dimorphism [76]. Therefore, the sexually dimorphic pain central sensitization mechanism of GT1b-induced pain is distinct from that of LPS-induced pain central sensitization.

We reported that GT1b administration led to IL-1β expression in the spinal cord, which contributed to pain sensitization [75]. IL-1β is initially expressed in its precursor form pro-IL-1β and then released upon cleavage by an inflammasome containing caspase-1 [77,78]. Our study found that GT1b administration induced inflammasome activation in the spinal cord in a sex-dependent manner; specifically, it induced caspase-1 expression only in male mice but not female ones. To investigate whether female sex hormones affect microglial inflammasome activation upon GT1b administration, we tested the effects of estrogen on inflammasome activation in primary glia [76]. Ultimately, estrogen did not inhibit GT1b-induced caspase-1 expression in vitro, so it is likely that estrogen may indirectly inhibit glial caspase-1 transcription in vivo. Although further studies are needed to elucidate the mechanism behind estrogen regulation of inflammasome activation in vivo, our study revealed a modulatory role of estrogen in GT1b-induced spinal cord inflammasome activation and subsequent pain central sensitization in females (Figure 1).

Numerous studies suggest the potential contribution of sex hormones to pain central sensitization. TRP channels, molecular sensors of harmful chemical and physical stimuli [79,80], are actively involved in the pathophysiology and sexual dimorphism of pain [81,82,83]. TRP channels are localized primarily in peripheral and central nerve terminals of DRG and trigeminal ganglia neurons [84]. Studies demonstrated the co-expression of estrogen receptor and TRPV1 channels in sensory neurons [85]. It was also reported that high levels of 17β-estradiol led to a lower threshold of mechanical and thermal sensitization while TRPV1 knock-out abrogated this nociceptive sensitization, implying the link between the TRPV1 channel and estrogen receptor in vivo [86]. For instance, although both sexes elicit sensitivity to capsaicin, an exogenous agonist of TRPV1, males required four-fold higher dose of capsaicin than females to evoke similar response [87]. Besides TRPV1, it is noteworthy to mention that testosterone also functions as a regulator for TRPM8, a TRP channel responsible for cold nociception [88].

Depletion of estrogen via ovariectomy resulted in alteration of the microglial transcriptome and an increased inflammatory phenotype. Estrogen depletion in females elicits greater pain hypersensitivity and upregulated inflammatory genes [89,90]. Microglia physiologically express steroid hormone receptors, rendering them sensitive to estrogen [91]. A recent study showed that estrogen administration significantly reduced allodynia and thermal hyperalgesia in patients with neuropathic pain after spinal cord injury. In addition, p38 and ERK MAPK activations in microglia were decreased by estrogen, inhibiting microglial activation [92]. It was also reported that estrogen alters pain sensitivity by blocking NLRP3 inflammasome formation and reducing superoxide dismutase production and phagocytic activity [93,94]. Moreover, previous reports showed that estrogen has a neuroprotective role in various neurological disorders, including Alzheimer’s disease, Parkinson’s disease, stroke, and brain trauma [92,95]. Thus far, the exact contribution of estrogen in nerve injury-induced pain central sensitization in humans is yet to be elucidated. Moreover, the exact mechanism of immune system regulation by sex hormones is not completely understood. Still, studies so far provide insight into the sex hormone-based differences between male and female responses to nerve injuries and subsequent pain central sensitization.

## 5. Conclusions and Future Directions

Sexual dimorphism in pain central sensitization exists and has been one of the most increasingly demanded research topics in both clinical and preclinical studies. Sex differences in pain responses in the clinical field are readily observable, considering various factors such as sex hormones and age disparities. Similarly, preclinical studies have also demonstrated sexually dimorphic pain phenotypes in various animal models of chronic pain and have elucidated factors involved in the sexual dimorphism of pain central sensitization. According to these studies, the difference in activation phenotype of spinal cord microglia underlies the sexually dimorphic pain central sensitization mechanism. In this process, male and female sex hormones are differently involved depending on central sensitization-causing stimuli. For instance, estrogen is involved in the sexual dimorphism in GT1b-mediated central sensitization, while testosterone is implicated in LPS-mediated central sensitization. In comparison, females employ a microglia-independent pain central sensitization pathway that involves recruitment of T-cells to mediate pain hypersensitivity after nerve injury. The female sex hormone estrogen may have neuroprotective roles against injuries, as was demonstrated in GT1b-induced allodynia. 

Taken together, these findings suggest that recognition of the broad gap of sex differences in pain and its mechanisms paves a wider path to further explore and elucidate the molecular, cellular, and behavioral sexual dimorphisms in pain central sensitization. The resolution and understanding of these mechanisms will provide useful and important implications for the development of pharmacological agents and effective therapeutic strategies to address broader and more diverse populations in the future.

## Figures and Tables

**Figure 1 cells-12-02028-f001:**
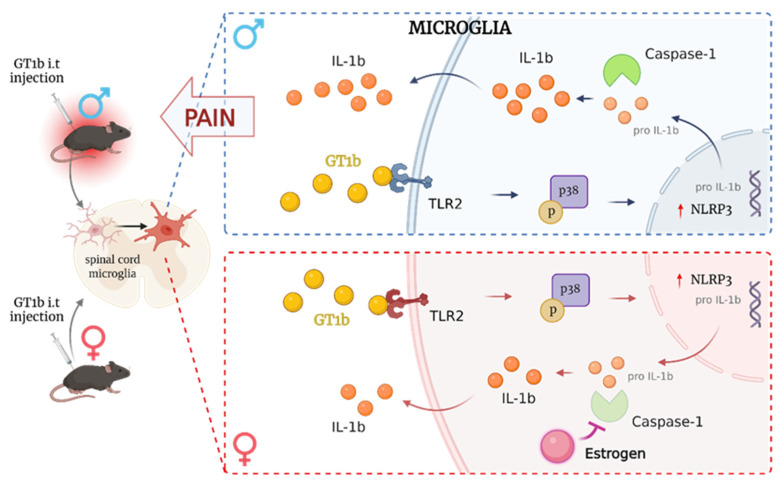
Sexual dimorphism in the mechanism of pain hypersensitivity induced by GT1b. The schematic illustration demonstrates male and female mice that were injected with GT1b in the spinal cord, with induced microglia activation and mechanical allodynia observed only in male mice (**left panel**). GT1b-induced pain hypersensitivity in male (**upper panel**) but not female mice (**lower panel**) evoked an increase in IL-1β expression through the activation of inflammasomes containing NLRP3 and caspase-1. Meanwhile, female mice showed no pain response and lower expression of IL-1β level, possibly due to indirect inhibition of estrogen to caspase-1.

## Data Availability

No new data were created or analyzed in this study. Data sharing is not applicable to this article.

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
