# Peer review of "Sexual Dimorphism in the Mechanism of Pain Central Sensitization"

_cells, 2023, doi:10.3390/cells12162028_

Round 1
Reviewer 1 Report
In the present study the Authors describe clinical and preclinical studies underlying sex differences in pain sensitization. The analysis is very interesting, and the manuscript is well written.
They include SNI, CCI and SNL as major models of chronic/ neuropathic pain. Have the Authors information about different chronic pain conditions, including metabolic disorders associated with chronic pain development (i.e. diabetic neuropathy) ? At this regards, differences between males and females have been recently reported in a model of chronic pain mediated by Vitamin D deficiency (Alessio et al., 2021 IJMS).
Minor editing of English language required
Author Response
We thank the reviewer for the insightful comment in including metabolic disorder neuropathies inducing chronic pain. We included the study of Alessio et al. on the sexual dimorphism in pain and microglial morphology of vitamin D-deficient male and female mice. The added information is in line 162-169 of the revised manuscript.
Reviewer 2 Report
The Manuscript highlight an important issue in the field of chronic pain pathophysiology.
The review article cover part of the literature focusing on the sexual dimorphism in immune-driven tactile allodynia. recent papers showing that also drugs for treating chronic diseases can work differently in male and female neuropathic mice (Boccella et al., FASEB J 2019). Very recent study also hihglighted a sexual dimorphism in TRP channels, assuming that it could be possible also a different nociception between male and female (Cabanero et al., Pharmacol Ther 2022). Several others important papers and review should be quoted by the Authors (Navratilova et al., Sci. Tranls. Med 2021; Coraggio et al., IJMS 2018 and others)
Several typos should be fixed within the manuscript
The english language need some style refinement
Author Response
We thank the reviewer for the insightful comments and suggestion to improve the manuscript. We have added these information and discussions accordingly as follows;
- We have mentioned the sexual dimorphism in drugs for treating chronic pain and referenced Boccella et al., 2019 FACEB J. We showed these at lines 35-36 of the revised manuscript.
- We also added discussions on sexual dimorphism in TRP channels underlying the differences in male and female nociception. These changes are added in lines 170-180 and lines 321-332 of the revised manuscript.
- Several other papers were referenced in the revised manuscript according to the suggestion of the reviewer.
- Manuscript errors were also checked and corrected accordingly.
Reviewer 3 Report
I really enjoyed reading the majority of this paper. I feel the explanations of the various animal pain models including SNI, SNL and CCI and the replication of Neuropathic Pain characteristics and the explanation of Central Sensitization really helped to support the discussions around the topic of Sexual Dimorphism . I have a few comments below that I would like the authors to consider. Overall I feel this is a good review paper but is missing a few things.
1. Please provide reference{s} to support statement in lines 27-30. In contrast to acute pain, which resolves when the tissue injury heals, chronic pain persists beyond the point of wound healing, leading to pathological changes in the peripheral and/or central nervous system.
2. Same for lines 30-31, it is important to have strong supporting references in the introduction and this is lacking thus far
3. Although this is a review paper, there should be a methods section describing the search strategy and why some articles are included while others are not. Some Narrative Reviews have even followed the methodological framework for scoping reviews outlined by the Preferred Reporting Items for Systematic Reviews and Meta Analyses Extension for Scoping Reviews (PRISMA-ScR), with obvious modifications.
Author Response
- We thank the reviewer for the helpful comments and suggestions. We provided references to support statements in line 27-30 describing acute and chronic pain; these statements are described in the studies of Melzack 1999 Pain, Basbaum et al., 2009 Cell, and Treede et al., 2019 Pain.
- To provide supporting references for lines 30-31, we cited studies from Benyamin et al. 2008 Pain Physician, Rusman et al. 2018 Curr. Rheumatol. Rep. In addition other references were included in the revised manuscript.
-
We thank the reviewer for the comment regarding the referencing strategy. Review articles published in MDPI do not strictly require a methodological framework for citing references. It must have been better to use PRISMA-ScR when we started working on the review, but we believe that the use of PRISMA-ScR is not necessary at current stage. However, we revised the manuscript and included more research and review papers supporting our statements and discussions. The references used are based on pertinent articles retrieved by a selective search from reputable journals.